# Towards Human-like Virtual Beings: Simulating Human Behavior in 3D Scenes

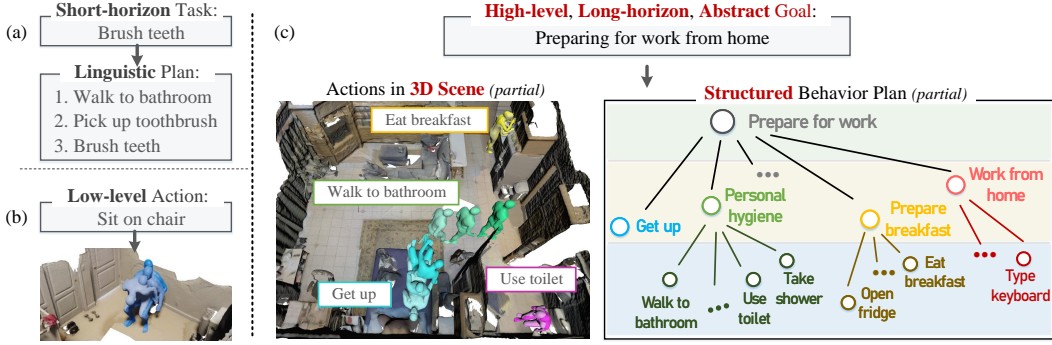

Figure 1: Previous research has primarily focused on: (a) linguistic-based short-horizon task planning, and (b) low-level human-scene interaction. (c) This study investigates the simulation of *high-level, long-horizon, abstract* goal-driven human behaviors in 3D scenes.

## Abstract

Building autonomous agents that can replicate human behavior in the realistic 3D world is a key step toward artificial general intelligence. This requires agents to be holistic goal achievers and to naturally adapt to environmental dynamics. In this work, we introduce ACTOR, an agent capable of performing *high-level, long-horizon, abstract* goals in 3D households, guided by its internal value similar to those of humans. ACTOR operates in a perceive-plan-act cycle, extending the ungrounded, scene-agnostic LLM controller with deliberate goal decomposition and decision-making through actively searching the behavior space, generating activity choices based on a hierarchical prior, and evaluating these choices using customizable value functions to determine the subsequent steps. Furthermore, we introduce BEHAVIORHUB, a large-scale human behavior simulation dataset in scene-aware, complicated tasks. Considering the unaffordable acquisition of human-authored 3D human behavior data, we construct BEHAVIORHUB by exploring the commonsense knowledge of LLMs learned from large corpora, and automatically aligning motion resources with 3D scene for knowledgeable generation. Extensive experiments on our established benchmark demonstrate that the proposed architecture leads to effective behavior planning and simulation. BEHAVIORHUB also proves beneficial for downstream task development. Our code and dataset will be publicly released.

## 1 Introduction

Building autonomous agents (*e.g.*, virtual beings, or humanoid robots) that can replicate human behavior in the realistic 3D world, has been a long-standing pursuit since the inception of AI (Diderot, 1911). The study could empower non-player game character (Riedl, 2012), underpin human-robot interaction (Riedl, 2019) and cooperation (Matsas & Vosniakos, 2017), populate virtual reality communities (Park et al., 2022), and accelerate Embodied AI (Puig et al., 2023).

Much progress has been made in vision and language models that imitate human motions (*cf*. Fig. 1 (b)) or propose linguistic plans (*cf*. Fig. 1 (a)). However, an effective 3D humanoid agent must go beyond by conquering three major barriers: *(i)* Holistic goal achievement from perception to action. As depicted in Fig. 1 (c), to accomplish a high-level goal (*e.g.*, '*prepare for work*'), the agent must process the perceived information (*e.g.*, '*lying on bed*'), decompose the goal into a series of activities (*e.g.*,'*get up*', '*use toilet*', '*eat breakfast*', *etc.*), and devise appropriate action plans for each; *(ii)* Environmental dynamics. Agents should be able to actively adjust their plans based on the environment, *e.g.*, determining if the bathroom is occupied when planning to *use toilet*; *(iii)* Vast and multifaceted human behavior space, where numerous viable paths exist to achieve even a single goal. For example, agents preparing for work can choose to *eat breakfast* before *using toilet* or vice versa. Also, when the bathroom is occupied, an intelligent agent can decide to whether *eat breakfast* first, or *continue waiting*, depending on its state and beliefs, *e.g.*, the desire to '*complete goal as soon as possible*' or '*save energy costs*'. A competent agent must process such value priorities to guide its selection and evaluation of actions and policies. Besides, the absence of a comprehensive testbed further poses great challenge for agent development and evaluation, constituting barrier *(iv)*.

In this work, we present ACTOR - a large language model (LLM) powered agent towards diligent simulation of human behavior in 3D realistic scenes (§4). ACTOR follows a perceive-plan-act cycle, addressing challenge *(i)* as envisioned. Using LLM as a central controller, it strategizes plans by searching the human behavior space. It actively maintains a tree of behaviors, where each node represents an intermediate step toward holistic goal achievement. The construction of this tree is guided by a hierarchical prior, *i.e.*, executable low-level actions are grouped into high-level semantic units, called activities, iteratively forming a hierarchical structure (*cf*. Fig. 1 (c)). The search progress is assessed using a set of customizable value functions that determine the likelihood of different intermediate candidates. In addition to being rational in common sense, ACTOR couples real-valued evaluations (*e.g.*, best *efficiency*) with personalized priors expressed through language commands (*e.g.*, description of *a neat person*). These outputs are converted into unified probabilities, allowing for the incorporation of the agent's characteristics and beliefs as value functions, thus addressing challenge *(iii)*. The planning process is dynamic and grounded in specific environmental values, empowering ACTOR to readjust its plans when faced with environmental changes or new language commands, effectively tackling challenge *(ii)*. This formulation facilitates the use of powerful search algorithms, *e.g.*, greedy search, beam search, and Monte Carlo tree search (MCTS), *etc*. In our experiments, we find MCTS exhibits superior performance compared to the others.

Furthermore, we establish a comprehensive environment for development and evaluation of agents like ACTOR, based on our newly proposed large-scale, scene-aware, behavior-rich dataset, dubbed BEHAVIORHUB (§5). One critical issue in constructing the human-authored benchmark is the high cost associated with acquiring and scaling high-quality, human-generated daily behavior data. Moreover, annotating large-scale behavior data further requires creativity in designing novel tasks and expertise in creating complete plans from scratch, which is also a challenging task for humans (Puig et al., 2018). Given this context, we propose to automatically synthesize 3D human behavior data by enhancing existing resources. Initially, we distill the tree-structured linguistic plans of human daily behaviors from LLM using in-context learning (Brown et al., 2020). Contactable objects and plausible interactions from the scanned 3D scenes (*e.g.*, ScanNet (Dai et al., 2017), *etc.*) and captured motion sequences (*e.g.*, AMASS (Mahmood et al., 2019), *etc.*) are attributed into the in-context prompt to ensure viable plans grounding in certain environment. Subsequently, we align the task-motion-scene data triplets by applying collision and contact constraints (Yi et al., 2022) for valid translation and rotation parameters. To promote diversity, we sample multiple plausible motion sequences for each high-level goal. In total, BEHAVIORHUB contains more than 1k daily goals over 10k high-quality behavior samples of 15.7 steps on average covering 1.5k scenes, that establishes a comprehensive testbed, that addresses the barrier *(iv)*.

We conduct extensive evaluations of ACTOR in §6. First, we find that ACTOR produces admissible plans and generates plausible motion sequences to simulate abstract, temporally-extended human behaviors. It outperforms the strong baselines by nearly doubling the overall success rate. This result is further confirmed by human evaluation. Then, we conduct several ablative experiments that limit ACTOR's access to each core design for thorough assessment. Finally, experiments on scene-aware and language-conditioned human motion generation demonstrate how BEHAVIORHUB can benefit the development of downstream task models.

## 2 RELATED WORK

**Human Behavior Simulation.** Simulating human behavior in realistic, open world environment like the one we inhabit is a long-standing topic in artificial intelligence (Bates, 1994). Historically, the topic has primarily been studied in the game worlds, focusing on enhancing player experiences through intelligent non-player game characters (NPCs) (Zubek, 2002; Aylett, 1999; Brenner, 2010). Early approaches rely on rule-based approaches like finite-state machines (Siu et al., 2021) and behavior trees (Colledanchise & Ögren, 2018). They provide a brute force way of manually crafting agent's behaviors, but cannot perform new procedures that were not hard-coded in their script (Umarov et al., 2012), limiting the generalizability. Another strand (Berner et al., 2019; Vinyals et al., 2019) involves using reinforcement learning, where agents learn its own policy through optimizing the learning algorithm on readily definable rewards over fixed task space, which is often limited to non-open, adversarial games, or blocks worlds only (Hausknecht et al., 2020; Miyashita et al., 2017; Tessler et al., 2017). The third line of research, represented by some pioneer works (Laird, 2001; Choi et al., 2021; Langley et al., 2005) in computational cognition, aims to build machines that operate directly in perceive-plan-act cycles, encompassing the nature of autonomous agents as originally envisioned. This formulation holds potential generalizability to most, if not all, open-world contexts. However, these studies are typically restricted to simplified environments, such as first-person shooter games (Choi et al., 2021), or 2D gridworlds (*e.g.*, Generative Agents (Park et al., 2023)), and focus on a reduced range of behaviors. Our work falls in the vein of the third category, while pushing the frontier towards human simulation in realistic 3D environments.

Recently, beyond the game world, new trending on social embodied intelligence, such as assistive robots, draws attentions to simulation of human collaborators' behavior. One notable effort is VirtualHome (Puig et al., 2018), which studies high-level human activities as plain sequences of atomic actions. However, its formulation is inherently environment-agnostic, which has been treated as a purely linguistic procedural planning problem in subsequent studies (Lu et al., 2023b; Huang et al., 2022). Furthermore, the discrete action space built upon manually crafted procedural knowledge also makes it an insufficient testbed. In contrast, with a special focus on environment-aware simulation of contiguous daily behaviors, our work paves one solid step accelerating the development of social embodied intelligence.

**LLM as Planner.** Recent years have witnessed remarkable progress in LLMs, demonstrating their emerging capacity to break down complex tasks into more manageable sub-tasks and devise appropriate plans for each (Shen et al., 2023; Lu et al., 2023a). LLMs have been successfully applied to solving mathematical problems (Imani et al., 2023; Azerbayev et al., 2023), reasoning on commonsense (Li et al., 2022), planning robotics tasks (Liang et al., 2023; Brohan et al., 2022), and very recently, manipulating external APIs on a web scale, expanding their capabilities beyond text generation. Based on the observation, we posit that LLMs can serve as a crucial component in extending the perception-decision-action space to construct human-like agents in realistic 3D environments. However, a crucial challenge remains: LLMs lack experience and interaction with their environment (Brown et al., 2020; Chowdhery et al., 2022), preventing them from ordering actionable and rational plans. This paper addresses this issue by incorporating customizable value functions into LLMs to evaluate and prioritize plans. The idea bears some resemblance to recent robotics research (Brohan et al., 2023; Huang et al., 2023). However, those methods necessitate retraining for every new set of primitive robotic skills, making them impractical for the complex and undefined human action space, which is difficult to define in advance. In contrast, we support plan evaluation using real-numbered functions and language-based rules without resource-intensive retraining.

**LLM as Data Generator.** Being trained on the large corpora of human-produced language, LLMs are believed to contain a wealth of information about the world (Li et al., 2021; Roberts et al., 2020). Given a handful of task-specific prompts, LLMs can generalize and generate more linguistic data in the same format, with the application of generating tabular data (Borisov et al., 2023), relation triplets (Chia et al., 2022), sentence pairs (Schick & Schütze, 2021), instruction data (Wang et al., 2023), *etc*. The idea seems naturally to be borrowed for human behavior data acquisition, where the requested human motion and daily activity procedure were commonly crowdsourced with high expense and complexity (Puig et al., 2018), limiting the scale and coverage of related datasets (Hassan et al., 2019). However, the process is non-trivial. The generated data is often blamed for low quality and diversity issues (Zhang et al., 2020; West et al., 2022). The 3D environment-aware nature of the task also poses unique challenges. To respond, we explore attributed prompts specifically

conditioned on the environment that not only mitigates the problems of low informativeness and redundancy, but also offers an effective workflow that can further empower other related domains, such as human-scene interaction (Wang et al., 2022b).

# 3 ENVIRONMENT

Simulating open-ended goals that resemble naturalistic human behaviors necessitates an environment capable of facilitating diverse agent affordances and interactions. Before delving into the detailed agent architecture, we first describe such environment we tailored for agents to instantiate in.

**Environmental Setup.** The environment features common affordances in a household, including:
- *Scene* of 3D textured meshes, that spans a house of multiple functional areas (*e.g.*, kitchen, *etc.*).
- *Objects* embedded in the scene (*e.g.*, stove in kitchen, bed in bedroom), with each constructed of 3D textured geometry and corresponding object state (*e.g.*, fridge: *opened*).
- *Humanoid agent(s)* defined by SMPL-X (Pavlakos et al., 2019), an expressive 3D human model of shape and pose of both body and hand. Agents reside in and interact with the scene and can influence the state of objects by their actions (*e.g.*, fridge: *opened* → *closed*).

As a start point, we demonstrate the environment of indoor household, keep the general formulation open for other environments, such as outdoor streets (Dai et al., 2022).

**Environmental Simulator.** Interactions within the environment are driven by the simulator of two components: *(i)* an engine that manages and evolves the states of objects in the environment; and *(ii)* a renderer that supports generation of multiple perceptual observations (*e.g.*, RGB, depth, 3D surfaces) for agents. At each time step, the simulator dynamically updates environment and collects egocentric, surround, or third-person-view information based on needs. We build our simulator upon Habitat-Sim (Savva et al., 2019; Szot et al., 2021) to ensure efficient and parallelizable simulation.

# 4 ACTOR AGENT

We aim to build humanoid agents that naturally simulate 3D human behaviors to complete daily goals, which can be lengthy, abstract, or ambiguous. Agents receive high-level goal described in language and are tasked with generating plausible motion sequences that align with the given scene.

## 4.1 AGENT ARCHITECTURE

Fig. 2 shows the workflow of ACTOR. The agent operates in a perceive-plan-act loop, using language as the generic interface connecting the three phases. At each step, it selectively perceives the world based on the target, transforming information into an environment description using heuristics. The observed information assists in decision-making through the LLM core, ultimately resulting in actions in the form of 3D motion sequences based on linguistic plans within the environment.

**Perception.** Our definition of perception extends beyond gathering information, such as 2D images or 3D point clouds, through the simulator (*cf*. §3). It encompasses attaining a deep understanding of the environment, including object properties, spatial relationships, and scene layouts, *etc*. In our preliminary implementation, information is perceived using readily available models and converted into linguistic descriptions using heuristic functions. More specifically, they take scene geometry, which includes object segmentation, as well as the agent's state of position and action type as input (Fig. 2 Right) and gives a linguistic description of the entire scene and the agent's surroundings as output. To provide a concrete example, a linguistic environmental description could be as follows:

ENVIRONMENT: {residential interior}; OBJECTS: {bed, desk, chair, kitchen counter, sink, television, ..., sofa}; SURROUNDINGS: {sink: *empty*, faucet: *turned on*, toilet: *vacant*}

The "Environment" and "Objects" fields offer the agent a comprehensive understanding of the human behaviors that may occur in the current scene. On the other hand, the "Surroundings" field provides the agent with information about interactive objects and their respective states in the surrounding.

**Plan.** At the core of ACTOR, a planning module receives the current environment and past behavior trial as input, generating action description as output. For the executable actions, it provides descrip-

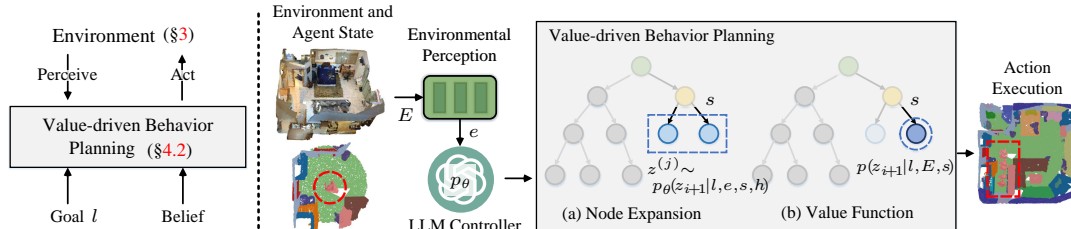

Figure 2: *Left*: Overall agent architecture (§4.1); *Right*: One-step in perceive-plan-act loop with value-driven behavior planning (§4.2).

tions of motion sequences. In the case of high-level activity, it bypasses the action stage and proceeds to break down the target further. We will provide a detailed explanation of this process in §4.2.

**Action.** For action, we specifically consider whole-body human actions in 3D scenes to closely resemble human behavior using the off-the-shelf models. It generates whole-body motion based on text and trajectory (Karunratanakul et al., 2023). For object-interactive actions, we further refine hand grasping using an isolated grasp estimation model (Taheri et al., 2020). Additionally, for moving actions like *walking*, trajectory paths are pre-estimated (Wang et al., 2022a). Detailed trajectory estimation process is provided in *supp.* §A.2

## 4.2 VALUE-DRIVEN BEHAVIOR PLANNING

We now delve into the details of planning phase. An LLM, denoted as $p_\theta$ with parameters $\theta$, functions as a controller to iteratively decompose the long-term goal $l$ described in language texts into shorter steps of behaviors based on the environmental dynamics $E$ and the corresponding perceived description $e$. For brevity, we define $E$ to include both the environment state and agent's state. We denote linguistic descriptions of behaviors as $z$. The problem is formulated as a search over a tree, where each node represents a state $s = \{z_{1\ldots i}\}$, representing a partial trial with the input and the sequence of behaviors taken thus far. The instantiation consists of three components: *(i)* Node expansion, which involves generating $k$ candidate behaviors for the next step search; *(ii)* Value functions to evaluate each node; *(iii)* Search algorithm that accounts for branch selection.

**Node Expansion.** The node is expanded through sampling from LLM with a window size $w$: $z^{(j)} \sim p_\theta(z_{i+1} \mid l, e, s, h)$, where $j \in \{1, \cdots, w\}$. We omit the basis prompting parts for brevity, which is also the case in the subsequent context. Here, we introduce a hierarchical heuristic, denoted as $h$, which is implemented through prompt instructions and demonstrations of specific actions and activity cases. This heuristic ensures that during each expansion, all candidates at the same level are restricted to executable actions or high-level semantic units of activities, that provides a more nuanced representation of behaviors for effectively modeling interchangeable activities.

**Value Function.** The value function assesses the state by considering the degree to which a specific behavior contributes to the achievement of the target, conditioned on the agents' beliefs reflecting their value. The search algorithm uses the output to determine which nodes to explore further and in what order. The likelihood is given by $p(z_{i+1} \mid l, E, s) \propto p_\theta(z_{i+1} \mid l, e, s) \cdot p_v(z_{i+1} \mid l, E, s)$. Here, the LLM provides us with $p_\theta(z_{i+1} \mid l, e, s)$, which represents the likelihood, based on commonsense, that a textual behavior is a valid next step. However, the LLM struggles to generalize or make inferences in the real environment since it is not grounded. On the other hand, $p_v$ provides the likelihood of the behavior being plausible in the current state of both the environment and agent, according to the defined values. We consider a set of values $\{p_{v_n}\}_n$ categorized into two types, and $p_v(z_{i+1} \mid l, E, s) = \prod_n p_{v_n}(z_{i+1} \mid l, E, s)$:

- *Real-valued function* assigns real values as outputs, such as the *shortest path* value, for which the function estimates the distance for each candidate action. The outputs of real-valued functions can be directly normalized into probabilities.
- *Language-based command* is implemented by prompting the LM with a value prompt that conveys the meaning of '*How likely is it for someone who is* `a neat person` *to take the following action,*

*considering ...*'. It reasons about the trial to generate a classification value of *sure/more-likely/less-likely/impossible*, which is empirically converted into probabilities set as $1.0/0.7/0.3/0.01$.

Values are also conditioned on the state of the environment and agent. This approach allows for reacting to environmental dynamics in addition to the active planning process.

**Search Algorithms.** Owing to the active planning process we have formulated, it is feasible to use different search algorithms based on the tree structure. We explore greedy search (Feo & Resende, 1995), beam search (Freitag & Al-Onaizan, 2017) and MCTS (Coulom, 2006), and evaluate their performance in experiments (*cf*. §6.2) where we find MCTS performs best. We use MCTS by default.

### 4.3 IMPLEMENTATION DETAIL

We employ GPT-4 and GPT-3.5-turbo API provided by OpenAI as the base LLMs. By default, we prompt the LLMs with four in-context examples as demonstrations. We set decoding temperature to 0 for more deterministic generation. We use official releases of conditional motion generation models and fine-tune them on BEHAVIORHUB using the default parameters for each to promote finer generation quality. By default, we set the window size to $w = 5$ and normalize the shortest path value based on the maximum rollout depth of the search algorithm, which is set to 3. In practice, we find one-time sampling is sufficient for effective tree search, eliminating the need for multiple samplings for branch aggregation.

**Reproducibility.** Our algorithm is implemented in PyTorch and LangChain. All experiments are conducted on Tesla A40 GPUs. Our code will be released for reproducibility.

## 5 BEHAVIORHUB BENCHMARK

To lay a solid foundation for future research, we build a large repository of common behaviors performed in daily household scenarios. Each sample contains three components: *(i)* a high-level goal (*i.e.*, root node); *(ii)* a tree-structured linguistic plan covering necessary intermediate-level steps (*i.e.*, intermediate nodes) and low-level steps (*i.e.*, leaf nodes) required to accomplish the goal; and *(iii)* scene-conditioned human motions corresponding to each executable step at either intermediate or low level. We illustrate one example in Fig. 3 (a). Intermediate step set can be empty. Both goal and steps in plans are human activities described with either concrete or abstract language (*e.g.*, 'go to sleep on bed' or 'feel tired'). To generate diverse and high-quality data, we employ a two-step pipeline that utilizes an LLM pretrained on extensive corpora: First, automatic generation of linguistic daily plans (§5.1); Second, alignment of the plans with 3D motions and scenes (§5.2). The entire data generation process is shown in Fig. 3 (b).

### 5.1 LINGUISTIC GOAL-PLAN GENERATION

The pipeline consists of three steps: *(i)* generate potentially incomplete goal-plan trees; *(ii)* complete and refine each tree; and *(iii)* filter out low-quality data. We provide detailed prompts in *supp.* §A.1.

**Attributed Goal-Plan Tree Generation.** In the initial step, we generate new goal-plan trees using a bootstrapping approach based on a small set of seed human-written samples. To ensure broader coverage and facilitate later alignment with specific scenes, we attribute the starting room, candidate objects and actions that interact with the activity in the plans into prompt demonstrations. We start the sample pool with 292 activities from ActivityPrograms (Puig et al., 2018), and consider 21 room types (*e.g.*, bathroom, gym), $\sim 10^3$ object types (*e.g.*, TV, table), and $\sim 10^2$ action types (*e.g.*, open, eat). Refer to the *supp.* §A.4 for the full list. For each step, we sample eight goal-plans as in-context examples from the pool while restricting the outputs to one room and ten types of objects. Out of the eight plans, six are from the human-written plans, and two are generated by the model in previous steps to enhance diversity.

The generated trees are then labeled with sequential order for intermediate nodes in the trees, which encompass multiple sub-step leaf nodes of individual actions. This sequential order is represented as *interchangeable groups*, where nodes within the same set are interchangeable with each other. We prompt the LM in a few-shot way to determine this.

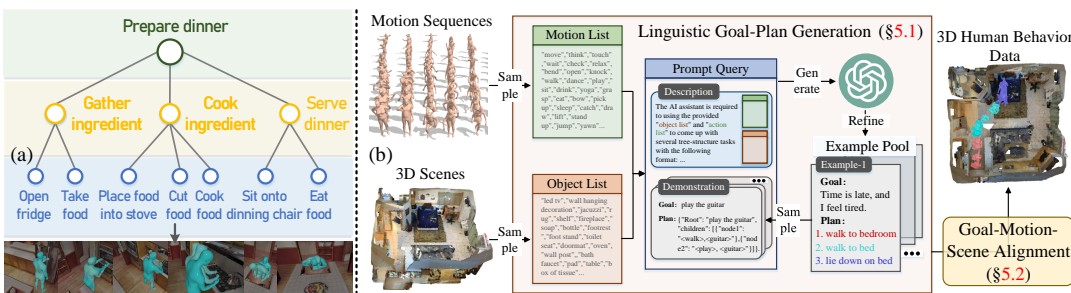

Figure 3: (a) An illustrative example of our BEHAVIORHUB dataset; (b) Semi-automatic 3D human behavior data generation pipeline (§5).

**Goal-Plan Tree Refinement.** We further improve the constructed trees by addressing two key aspects: *(i)* we complete any missing internal plan steps, which can often be revised based on commonsense, *e.g.*, opened the fridge without closing it. *(ii)* we enhance the root node descriptions to be more abstract, *e.g.*, transforming '*use toilet*' to '*feel the call of nature*'. We accomplish both aspects by querying the LLM for suggestions.

**Filtering.** To promote diversity, a new tree is added to the pool only if its BERTScore similarity (Zhang et al., 2019) with any existing goal-plan tree is below 0.5. Additionally, we utilize LLM to assess the generated trees by asking the question, '*Is this a valid plan?*' Any plans flagged as '*invalid*' are filtered out to ensure high-quality data.

## 5.2 GOAL-MOTION-SCENE ALIGNMENT

Once the goal-plan trees are formed, we proceed to ground them in the 3D environment. We propose to synthesize 3D behaviors by leveraging existing resources of human motions (*i.e.*, AMASS (Mahmood et al., 2019), BABEL (Punnakkal et al., 2021), GRAB (Taheri et al., 2020)), and indoor scenes (*i.e.*, ScanNet (Dai et al., 2017), HM3D (Ramakrishnan et al., 2021), Replica (Straub et al., 2019)). For each tree, we first sample actions and objects from the resources using corresponding labels to fulfill the executable activities in forms of combinations of actions on objects. Then, given the sampled motion, we aim to generate plausible and contiguous interactions to the sampled object in the scene.

**Motion Alignment.** We first put motion clips into the scene anchored by the contactable objects. We optimize the translation and rotation parameters matrices by minimizing the *collision* and *contact* losses from MOVER (Yi et al., 2022). A unified SDF volume is calculated, and all contact vertices for all frames are accumulated in 3D space. The motion is aligned through a transformation of the two matrices. This joint optimization improves human-object contact and resolves 3D interpenetrations between humans and the scene.

**Sequence Blending.** The aligned motion sequence, which may be sparse and not spatially connected, is blended using a Transformer-based motion completion method (Duan et al., 2021). This results in a contiguous motion sequence that aligns with both the scene and goal plan. In practice, we find a single network is capable of delivering satisfactory results that align with prior research (Kim et al., 2022; Shafir et al., 2024).

**Verification.** Finally, to ensure the data quality, each example is examined by three verifiers who vote on whether the plan (motion) is complete (valid). If an example receives majority approval, it is accepted; otherwise, it is dropped.

## 5.3 DATASET STATISTICS

Our dataset consists of 10k human behavior samples in 1.5k 3D scenes, covers 2k unique activity over 0.1k actions and 1k objects, which is an order of magnitude larger than the human-authoring ActivityPrograms (Dai et al., 2017). On average, each high-level goal has 15.7 steps, resulting in a total of 10.1 corresponding motion sequences of 83.3 frames, that span over 8.6M motion frames. Each activity corresponds to 4.1 different motion sequences in 1.7 rooms. Owing to the automatic

Table 1: **Quantitative results** on `Main` and `Dynamic` set of BEHAVIORHUB. '↓' indicates smaller values are better. See §6.1 for details.

| | Method | Behavior Planning | | Behavior Simulation | | | | |
|---|---|---|---|---|---|---|---|---|
| | | S-BLEU | BERT-S | SSR | GSR | GSRPL | FID ↓ | Accuracy ↑ |
| Main Set | LLMaP (Huang et al., 2022)[ICML2022] | 0.089 | 0.825 | - | - | - | - | - |
| | HuggingGPT (Shen et al., 2023)[NeurIPS23] | 0.132 | 0.856 | 0.533 | 0.317 | 0.161 | 5.386 | 0.620 |
| | ACTOR (**Ours**) | **0.170** | **0.879** | **0.601** | **0.472** | **0.351** | **2.087** | **0.773** |
| Dynamic Subset | Human | 0.203 | 0.959 | - | - | - | - | - |
| | LLMaP (Huang et al., 2022)[ICML2022] | 0.069 | 0.821 | - | - | - | - | - |
| | HuggingGPT (Shen et al., 2023)[NeurIPS23] | 0.099 | 0.830 | 0.407 | 0.164 | 0.073 | 9.116 | 0.505 |
| | ACTOR (**Ours**) | **0.135** | **0.862** | **0.515** | **0.306** | **0.212** | **3.141** | **0.697** |

data generation pipeline, our dataset achieves even *higher diversity and coverage with low human cost*. More analyses are provided in *supp.* §A.4.

## 5.4 EVALUATION METRIC

Our evaluation of the plausible behavior simulation encompasses two key aspects: *(i)* the simulation should generate linguistically reasonable behavior plans (*i.e.*, behavior planning); and *(ii)* align them with natural motion sequences within the 3D environment (*i.e.*, behavior simulation).

- For behavior planning, Sentence-BLEU (Papineni et al., 2002), and BERTScore (Zhang et al., 2019) are used to measure the semantic similarity between the ground-truth plans and predictions. We report maximum scores attained across gt plan variants to ensure the evaluation be order-invariant *w.r.t.* interchangeable sub-steps.
- For behavior simulation, we consider three sets of metrics: *(i) Success Rate*: The step success rate (SSR) records the percentage of steps where the agent successfully completes the step objective, defined by a contact distance threshold. For example, we consider lying down to be successful if both the hip and head of the humanoid are within 30 cm of the target location (Hassan et al., 2023). The goal success rate (GSR) is measured to determine whether all steps in the entire plan are successfully executed; *(ii) Goal Success Rate Weighted by Path Length* (GSRPL): It judges how efficient was the agent at finishing the goal, defined as $GSR \cdot \frac{g}{\max(g,l)}$. Here $g$ is ground-truth path length and $l$ is the agent's path length; *(iii) Motion Quality*: We further evaluate the overall quality of generated motions using Frechet Inception Distance (FID) and recognition accuracy measured with the final layer of a pretrained standard RNN action recognition classifier as motion feature extractor, which offers intuitive and fine-grained assessments of generation quality.

## 6 EXPERIMENT

### 6.1 PERFORMANCE ON BEHAVIORHUB BENCHMARK

**Dataset Split and Dynamic Subset.** We randomly sample 200 held-out goals as demonstration set from which we select example(s) for prompting language models, and also as training data for fine-tuning conditional generation models. The remaining ones are used for evaluation.

To thoroughly examine the capacity of benchmarked models in managing environmental dynamics, we manually create a subset of 300 samples from the original BEHAVIORHUB, called `Dynamic` subset. In each sample, we carefully configure the agent's environment state-aware triggers to guarantee a distinct goal-plan solution for a specific goal, *e.g.*, we designate the bathroom as occupied only after certain pre-request steps have been fulfilled; or incorporate language commands that specify agent characteristics, further contributing to a unique goal-plan preference.

**Competitors.** We benchmark two top-leading LLM-based models: LLMaP (Huang et al., 2022), a procedural planning model, and HuggingGPT (Shen et al., 2023), a general tool agent, to probe the human behavior simulation ability in existing techniques. LLMaP is designed to operate solely on textual inputs, and lacks the capability to perform behavior simulation. Therefore, our evaluation and report focus solely on its behavior planning ability.

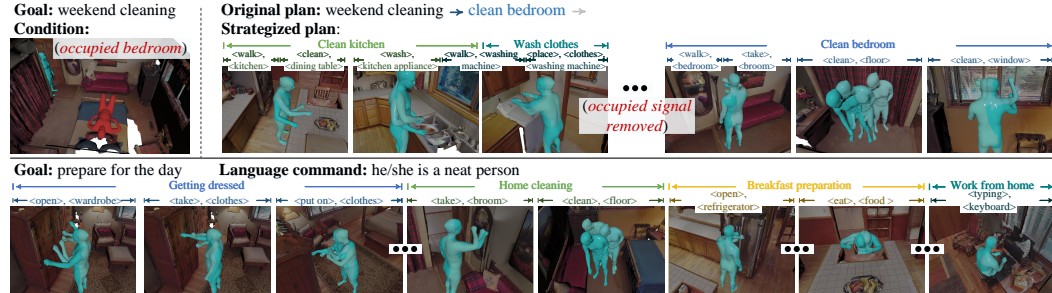

Figure 4: **Qualitative results** of ACTOR on the challenging BEHAVIORHUB Dynamic subset. Please refer to §6.1 for more details.

**Quantitative Evaluation.** Table 1 summarizes the automatic evaluation results, from which we take three major observations: *(i)* ACTOR achieves the best results on both behavior planning and simulation and outperforms the strong baseline by nearly doubling the GSR, indicating the effectiveness of our value-driven behavior planning design. *(ii)* On the challenging Dynamic subset, both LLMaP and HuggingGPT experience a noticeable decline in terms of BERT-S, whereas our ACTOR performs much better. To illustrate, in the case shown in Fig. 4, both LLMaP and HuggingGPT directly ignore the occupied signal and give an irrational plan of heading directly to the bedroom, causing a failure. This observation confirms our claim that the ungrounded LLM falls short in terms of environment-aware behavior planning. *(iii)* Although our ACTOR shows more promising results, there still remains a significant gap compared to human performance. This highlights the need for developing more sophisticated behavior simulation models.

**Human Evaluation.** For a comprehensive evaluation, we engage five participants as human evaluators. They rate the model performance based on three aspects: *(i)* Completeness: assess whether the motion steps can successfully complete the target goal, capturing semantic completeness; *(ii)* Ra-

Table 2: **Human-evaluated results** on Dynamic subset.

| Method | Complete. | Rational. | Quality |
|---|---|---|---|
| Human | 4.02 | 4.85 | - |
| LLMaP [ICML2022] | 2.23 | 2.48 | - |
| HuggingGPT [NeurIPS23] | 2.71 | 2.89 | 3.18 |
| ACTOR (**Ours**) | **3.05** | **3.47** | **3.75** |

tionality: evaluate whether the sequence includes necessary steps in the correct sequential order to accomplish the target goal, capturing sequential order correctness; and *(iii)* Quality: reflect the naturalness and smoothness of the motion sequence, capturing motion quality. The results of the human ratings, based on a 5-point Likert scale, are reported in Table 2. The human subjective judgments generally align with the trends reflected by Table 2, confirming the reliability of our constructed automatic evaluation framework. Also, the results reaffirm that ACTOR yields plausible behavior simulation. However, it is worth noting that human-written plans are consistently preferred over our results, underscoring the challenging nature of our newly proposed BEHAVIORHUB benchmark.

**Qualitative Analysis.** Examples from ACTOR on BEHAVIORHUB Dynamic are visualized in Fig. 4. In addition to simulating behavior to strategize and successfully achieve the desired goal, ACTOR is able to respond to environmental changes and adapt to language commands. For instance, to accomplish the goal of '*weekend cleaning*', while someone is still in bed, *i.e.*, the bedroom is occupied, the agent prioritizes scheduling the kitchen first and then the bedroom, waiting for the person to wake up. In the second example, considering the truth that someone is *a neat person*, he/she engages in *home cleaning* before *eating breakfast*.

## 6.2 DIAGNOSTIC EXPERIMENTS

A set of ablative studies is conducted on BEHAVIORHUB Dynamic for indepth analyzing each component in ACTOR, using BERT-S, GSR, GSRPL (*cf*. §5.4) as evaluation metrics

**Key Component Analysis.** We first validate the importance of our proposed components by attaching them one at a time in Table 4b. The 1st row reports the result of a bare baseline model, which produces a global plan based on the given linguistic goal. Next, in the 2nd row, we transition from one-pass planning to active tree search, resulting in improved performance and supporting our claim that active

Table 3: **Ablative experiments** on the `Dynamic` subset of our proposed BEHAVIORHUB. Please refer to §6.2 for more details.

(a) Key Component Analysis

| Method | BERT-S | GSR | GSRPL |
|---|---|---|---|
| Baseline | 0.811 | 0.140 | 0.062 |
| + Active Search | 0.837 | 0.235 | 0.132 |
| + Hier. Prior | 0.849 | 0.261 | 0.154 |
| + Value Func. | **0.862** | **0.306** | **0.212** |

(b) Search Algorithm

| Algorithm | BERT-S | GSR | GSRPL |
|---|---|---|---|
| Greedy | 0.840 | 0.244 | 0.151 |
| Beam | 0.853 | 0.287 | 0.186 |
| MCTS | **0.862** | **0.306** | **0.212** |

(c) Modular Scalability

| LLM | BERT-S | GSR | GSRPL |
|---|---|---|---|
| Vicuna-7b | 0.808 | 0.092 | 0.063 |
| GPT-3.5 | 0.833 | 0.176 | 0.116 |
| GPT-4 | **0.862** | **0.306** | **0.212** |

Table 4: **Quantitative results** (§6.3) on two downstream tasks. We use '†' to indicate using BEHAVIORHUB for additional training.

(a) Scene-aware Motion Generation

| Method | MPJPE↓ | MPVPE↓ |
|---|---|---|
| (Wang et al., 2021) [CVPR21] | 242.50 | 222.13 |
| †(Wang et al., 2021) [CVPR21] | **201.56** | **189.21** |

(b) Language-conditioned Motion Generation

| Method | FID↓ | R Precision↑ |
|---|---|---|
| MDM (Tevet et al., 2023) [ICLR23] | 0.544 | 0.611 |
| †MDM (Tevet et al., 2023) [ICLR23] | **0.471** | **0.705** |

planning can mitigate the impact of environmental changes. Moreover, the $3^{rd}$ row gives the score when the hierarchical behavior structure prior is employed when spanning searching branches. As seen, this leads to moderate improvement by constraining the search space with interchangeable semantic units, highlighting its necessity in handling vast and complex human behavior space. Finally, as shown in the $4^{th}$ row, incorporating value function significantly enhances the overall success rate and success rate over path length, aligning with the rational preference for the shortest path.

**Search Algorithm.** Table 3b reveals the impact of search algorithms (§4.2), *i.e.*, greedy search, beam search, and MCTS, with window sizes of 5 for the later two algorithms. The default strategy, MCTS, shows optimal results, which aligns with the widely accepted understanding that MCTS is more effective when dealing with a large solution space.

**Modular Scalability.** In Table 3c, we investigate the modular scalability by employing various LLM cores, including GPT-3.5, GPT-4 and open-source LLM Vicuna-7b (Chiang et al., 2023). As the LLMs' capabilities improve, ACTOR exhibits a continuous enhancement in its behavior simulation performance. This observation substantiates the notion that our ACTOR system possesses great potential for accommodating the development of more powerful LLM cores. Moreover, this shows that BEHAVIORHUB, in conjunction with the human behavior planning task, serves as a robust testbed for evaluating the planning capability of LLMs.

## 6.3 BEHAVIORHUB FOR DOWNSTEAM TASK

We further probe the effectiveness of BEHAVIORHUB by incorporating it as additional training data for two downstream tasks: scene-aware (Wang et al., 2021) and language-conditioned motion generation (Tevet et al., 2023), which focus on generation without planning. We follow the official implementation of both methods and begin by pre-training the two models on BEHAVIORHUB. Subsequently, we fine-tune then evaluate these models on PROX (Hassan et al., 2019) and HumanML3D (Guo et al., 2022), respectively. The evaluation results are reported in Table 4. As observed, BEHAVIORHUB significantly enhances model performance across all evaluation metrics for both tasks, highlighting the potential of our created dataset in facilitating broader applications within the realm of motion generation.

## 7 CONCLUSION

We present ACTOR, an LLM-powered agent towards realistic simulation of human behavior in 3D scenes. ACTOR integrates an LLM controller to perform complex behavior through planning on goal decomposition guided by hierarchical activity prior. The value-driven mechanism further deepens its understanding of environment. We demonstrate its potential in the simulated 3D indoor environment constructed using our newly created large-scale, scene-aware, behavior-rich dataset, BEHAVIORHUB. Evaluations suggest effective behavior planning and simulation of ACTOR.

## REPRODUCIBILITY STATEMENT

We believe we have revealed sufficient details of data (§5), pipeline (§4), and running details (§4.3, *supp.* §A.2, *supp.* §A.3). All evaluation assets are publicly accessible, and we adhere to standard evaluation protocols for procedural planning (Puig et al., 2018; Huang et al., 2022) and human-scene interaction (Hassan et al., 2023; Wang et al., 2022b) to report and compare the results. The cited assets are listed in §5 and §6.3, with their licenses detailed in *supp.* §A.6. For additional assurance, we will ensure the public availability of the code, dataset creation instructions, agent implementation, and model checkpoint upon acceptance.

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

# A  APPENDIX

In the appendix, we provide the following items that shed deeper insight on our contributions:

- §A.1: Details about data generation prompts.
- §A.2: Details about motion trajectory generation and human-scene interaction.
- §A.3: Details about MCTS process.
- §A.4: More dataset statistics.
- §A.5: More qualitative visualization and detailed goal-plan json.
- §A.6: Discussion of legal/ethical considerations and limitations.

## A.1  PROMPTS FOR DATA GENERATION

We give full details of the prompts used in generating linguistic goal-plan trees, including in goal-plan trees initialization Table 5, attribution of interchangeable groups in Table 6, and goal-plan tree refinement in Table 7.

## A.2  MORE DETAILS OF MOTION TRAJECTORY GENERATION AND HUMAN-SCENE INTERACTION

In the action module of ACTOR, we generate whole-body human actions in 3D scenes using off-the-shelf conditional motion generation models. Here we provide more details on how we achieve motion trajectory generation and human-scene interaction.

For motion trajectory generation, once the linguistic planning step provides us with a parsed <action, object> pair, we categorize the action into two types: still and moving. First, for still actions, such as stand up and knock, no trajectory estimation is necessary as the human remains in a fixed position. Then, as we mentioned in §4.1, for moving actions like walking, trajectory paths are pre-estimated Wang et al. (2022a). We adapt the trajectory estimation module from Wang et al. (2022a). The end position is sampled based on contact and collision rules, taking into account the scene and targeted object geometry. The goal is to position the human close to the target while avoiding collisions with walls. The start position is determined based on the previous step's end position. Subsequently, this module utilizes an improved A* path search algorithm, considering the start-end position and the entire scene geometry, to generate the final trajectory.

Furthermore, for achieving human-scene interaction, we construct the leaf nodes of these goals as <scene, text, motion> pairs to finetune the conditional motion generation model Karunratanakul et al. (2023), where a scene-conditioned branch is added and implemented with a pretrained and fixed Point Transformer Zhao et al. (2021) to achieve human-scene interaction. The conditional motion generation model takes pre-estimated trajectory, text description, and scene geometry as input to generate the whole-body motion. While the grasp estimation model further refine the hand pose. During finetuning, we keep the hyperparameters consistent with the official implementation, except for using a learning rate that is half of the original value. This adjustment already yields moderate adaptation.

## A.3  MORE DETAILS OF MCTS PROCESS

In Fig. 2, we illustrate a generalized view of the tree structure used in different algorithms such as greedy search, DFS, BFS, A* search, and also MCTS. The value function assigns values to each node, while node expansion determines the probability of transitioning from the node state. In this section, we present a more detailed description of MCTS process based on the proposed value-driven planning approach. Specifically, in MCTS, the root node represents the current state of the system being executed. Each child node corresponds to a potential action or step that can be taken from the current state. These nodes have associated values and states, which include information about the current scene and the state of the human involved. First, the initial value of a node is determined by the value function (*cf*. §4.2). This value is then updated during the backpropagation phase. Second, during the expansion phase, new child nodes are created by sampling from LLM within the Node Expansion process. Third, in the backpropagation phase, the results of a two-step rollout are summarized. This involves considering the values of the two-step children and updating the value of the parent node

accordingly. Finally, the MCTS process continues to iterate until a termination condition or goal is reached, signifying that the search is finished.

### A.4 MORE DATASET STATISTICS

We provide lists of most frequently used motion and objects in Fig. 6-7 and Table 8-9. Example scene is illustrated in Fig. 5. We next give brief description of the scene dataset we incorporate for data generation: *(i)* ScanNet Dai et al. (2017) is a widely known dataset in computer vision and 3D scene understanding. ScanNet is a large-scale RGB-D dataset containing 3D scans of indoor spaces, along with detailed semantic and instance-level annotations. It is commonly used for tasks such as 3D scene understanding, object recognition, and semantic segmentation. Researchers and developers use ScanNet to train and evaluate algorithms for various applications related to understanding the 3D structure of indoor environments. *(ii)* Habitat-Matterport 3D Research Dataset (HM3D) Ramakrishnan et al. (2021) is the largest-ever dataset of 3D indoor spaces. It consists of 1,000 high-resolution 3D scans (or digital twins) of building-scale residential, commercial, and civic spaces generated from real-world environments. Researchers can use it with FAIR's Habitat simulator to train embodied agents, such as home robots and AI assistants, at scale.

### A.5 MORE VISUALIZATION AND GOAL-PLAN JSON

More illustrations of qualitative visualization and detailed goal-plan tree JSONs are given in Fig. 8.

### A.6 DISCUSSION

**Asset License and Consent.** We build BEHAVIORHUB on top of three human motion datasets (*i.e.*, AMASS Mahmood et al. (2019), BABEL Punnakkal et al. (2021), GRAB Taheri et al. (2020)), and two indoor scene datasets (*i.e.*, ScanNet Dai et al. (2017), HM3D Ramakrishnan et al. (2021)), that are all publicly and freely available for academic purposes. We implement our agent with LangChain codebase using GPT-3.5 and GPT-4 models. AMASS (https://amass.is.tue.mpg.de/) is released under this License; BABEL (https://babel.is.tue.mpg.de/) is released under this License; GRAB (https://grab.is.tue.mpg.de/) is released under this License; Scan-Net (http://www.scan-net.org/) is released under this License, and the code is released under the MIT license; HM3D (https://aihabitat.org/datasets/hm3d-semantics/) is released under this License; LangChain codebase (https://github.com/langchain-ai/langchain) is released under the MIT license. GPT models from OpenAI are available for academic research under this License.

**Crowdsourcing Data Collection.** BEHAVIORHUB is primarily collected through an automated data collection pipeline, with minimal human intervention required for verification. In addition, we conduct user studies to evaluate the quality of the human-subjective generation. All human experts involved in the annotation and evaluation process are well-informed that their contributions will be utilized for academic research, and their consent is obtained through signed agreements. To ensure privacy and equality, the annotation process strictly adheres to guidelines that prevent the disclosure of personal information about the experts and minimize data bias.

**Limitation Analysis.** One limitation of this work is that although the generated human motions are scene-aware, the interaction with objects is currently assumed to be static. In our future work, we aim to enhance the capabilities of BEHAVIORHUB and the ACTOR agent by incorporating interactions with interactive objects. To achieve this goal, we have developed our environment using the Habitat-Sim simulator, which offers the necessary flexibility to realistically simulate these interactions in future developments. Furthermore, we are committed to designing a more realistic benchmark and algorithm for simulating interactions, ensuring that our work aligns with future advancements in this area. To encourage broader exploration and engagement from the research community, we will also release our complete code implementation, comprising the environment simulator, dataset construction, and agent implementation.

**Broader Impact.** This study focuses on simulating high-level, long-horizon, abstract goal-driven human behaviors in 3D scenes. The approach has several positive implications, including advancements in Embodied AI, potential to populate virtual reality communities, and enhancement of non-player game character development. However, there are potential negative consequences to consider. The

generated results could be exploited for malicious purposes, such as the creation of highly realistic and deceptive virtual characters for social engineering or online scams. While this issue falls outside the scope of this paper, we intend to release our models in a gated manner to ensure that they are solely used for academic research purposes and prevent any misuse.

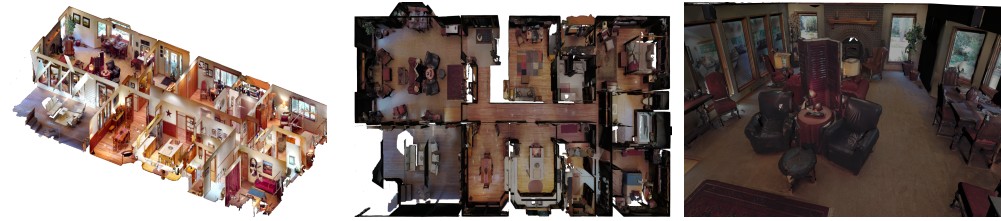

Figure 5: Example Scene. (a) Global view from a slanted perspective; (b) Global top-down view; (c) Local view of a living room.

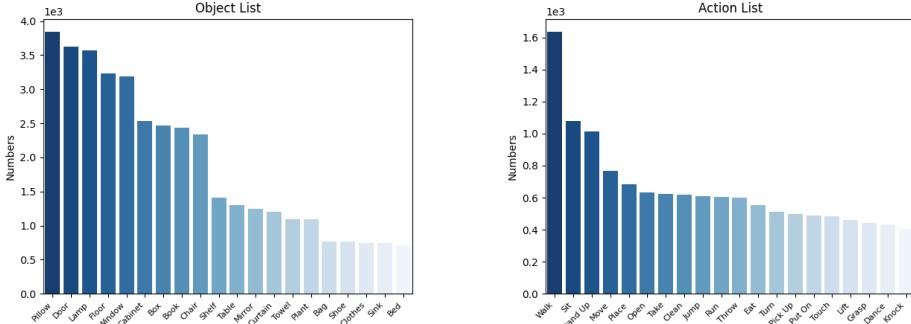

Figure 6: (a) Counts of actions in our BEHAVIORHUB dataset; (b) Object counts.

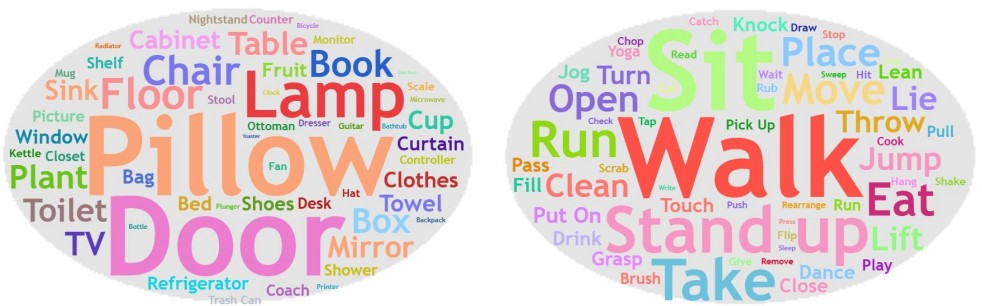

Figure 7: (a) Most frequently used objects; and (b) Most frequently used actions.

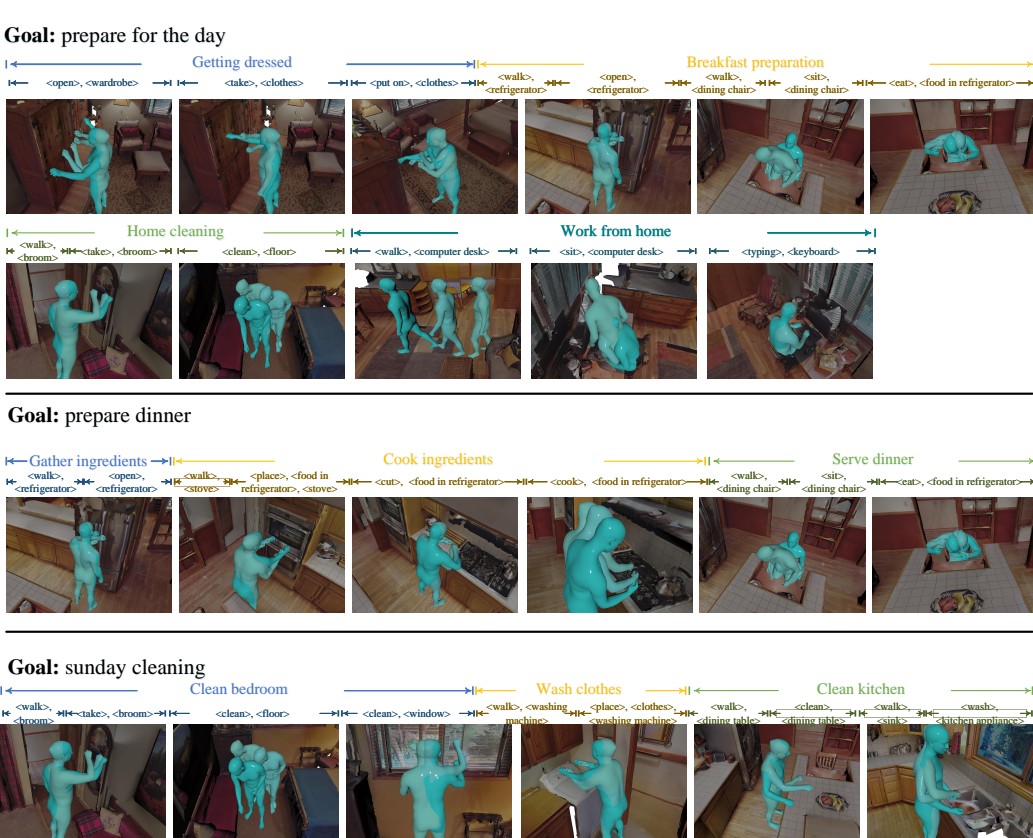

Figure 8: More qualitative visualization on our BEHAVIORHUB dataset.

Table 5: Detailed prompt design for goal-plan trees initialization.

| Prompt |
| --- |
| **#1 goal-plan trees Initialization Stage** - The following is a friendly conversation between a human and an AI. The AI is professional and can generate multiple goal-plan trees with lots of specific details from its context. The AI assistant is required to using the provided "object list" and "action list" to come up with several tree-structure tasks with the following format: [{"Root": task, "children": [{"node1": subtask, "children": [{"node1-1": ACTION, "children": []}, {"node1-2": ACTION, "children": []}]}, {"node2": subtask, "children": [{"node2-1": ACTION, "children": []}]}]}]. Note: "ACTION" must be "<action>", "<action>, <object>" or "<action>, <object>, <object>", non-leaf nodes must be "task" or "subtask". Intermediate nodes must be grouped, and the order of nodes in the same group is interchangeable. The AI assistant must reply in JSON format. The "task" or "subtask" field represents high-level task such as "Read a book", "Take a shower" or "Watch TV". The "task" or "subtask" must be complex activities or objectives in household. The "<action>" must be selected from the "action list", "<object>" must be selected from the "object list", and together they achieve the corresponding "task". Here are the "object list" and "action list" provided: {{Object List}}, {{Action List}}. To assist with goal-plan tree generation, here are several cases for your reference: {{Demonstrations}}. |

| Demonstrations |
| --- |
| **Now, please generate a tree-structure tasks:**
{"Root": "play the toy", "children": ["node1", "node2"], "interchangeable groups": ["node"]}
  {"node1": "walk toy", "children": [] }
  {"node2": "play toy", "children": [] } |
| **Now, please generate a tree-structure tasks with more branches and more depths. Remember you should reply in JSON format and the "<action>" must be selected from the "action list", "<object>" must be selected from the "object list":**
{"Root": "morning routine", "children": ["node1", "node2", "node3"]}
  {"node1": "have breakfast", "children": ["node1-1", "node1-2", "node1-3"] }
    {"node1-1": "<walk>", <refrigerator>", "children": []}
    {"node1-2: "<open>, <refrigerator>", "children": []}
    {"node1-3: "<garb>", <food in refrigerator>", "children": []}
  {"node2": "eating", "children": ["node2-1", "node2-2", "node2-3"] }
    {"node2-1": "<walk>, <dining table>", "children": [] }
    {"node2-2": "<sit>, <dining table>", "children": [] }
    {"node2-3": "<eat>", "children": [] }
  {"node3": "work", "children": ["node3-1", "node3-2", "node3-3"] }
    {"node3-1": "<walk>, <computer chair>", "children": [] }
    {"node3-2": "<sit>, <computer chair>", "children": [] }
    {"node3-3": "<typing>, <keyboard>", "children": [] } |

*(Row label: Goal-Plan Trees Initialization)*

Table 6: Detailed prompt design for intermediate nodes labeling.

| Prompt |
| --- |
| **#2 Intermediate Nodes Labeling Stage** – With the input goal-plan tree in JSON, the AI should assist in labeling the intermediate nodes in the trees using the attribute "interchangeable groups". Note: Intermediate nodes must be grouped, and the order of nodes in the same group is interchangeable. For a more comprehensive understanding of this procedural step, please refer to the corresponding demonstration {{Demonstrations}}. Remember you should reply in JSON format. |

| Demonstrations |
| --- |
| Please label the intermediate nodes in the following goal-plan tree:
**Query:**
{"Root": "evening routine", "children": ["node1", "node2", "node3"]}
  {"node1": "watch TV", "children": ["node1-1", "node1-2", "node1-3"] }
    {"node1-1": "<walk>", <couch>", "children": []}
    {"node1-2: "<sit>, <couch>", "children": []}
    {"node1-3: "<press>", <remote>", "children": []}
  {"node2": "have dinner", "children": ["node2-1", "node2-2", "node2-3", "node2-4", "node2-5", "node2-6"] }
    {"node2-1": "<walk>, <refrigerator>", "children": [] }
    {"node2-2": "<open>, <refrigerator>", "children": [] }
    {"node2-3": "<take>, <food in refrigerator>", "children": [] }
    {"node2-4": "<walk>, <dining chair>", "children": [] }
    {"node2-5": "<sit>, <dining chair>", "children": [] }
    {"node2-6": "<eat>, <food in refrigerator>", "children": [] }
  {"node3": "edtime routine", "children": ["node3-1", "node3-2"] }
    {"node3-1": "<walk>, <bed>", "children": [] }
    {"node3-2": "<lie>, <bed>", "children": [] }
**Response:**
{"Root": "evening routine", "children": ["node1", "node2", "node3"], "interchangeable groups": ["group1", "group2"]}
  {"node1": "watch TV", "children": ["node1-1", "node1-2", "node1-3"] }
    {"node1-1": "<walk>", <couch>", "children": []}
    {"node1-2: "<sit>, <couch>", "children": []}
    {"node1-3: "<press>", <remote>", "children": []}
  {"node2": "have dinner", "children": ["node2-1", "node2-2", "node2-3", "node2-4", "node2-5", "node2-6"] }
    {"node2-1": "<walk>, <refrigerator>", "children": [] }
    {"node2-2": "<open>, <refrigerator>", "children": [] }
    {"node2-3": "<take>, <food in refrigerator>", "children": [] }
    {"node2-4": "<walk>, <dining chair>", "children": [] }
    {"node2-5": "<sit>, <dining chair>", "children": [] }
    {"node2-6": "<eat>, <food in refrigerator>", "children": [] }
  {"node3": "watch TV", "children": ["node3-1", "node3-2"] }
    {"node3-1": "<walk>, <bed>", "children": [] }
    {"node3-2": "<lie>, <bed>", "children": [] }
  {"group1": [{"node1"}, {"node2"}]}
  {"group2":[{"node3"}]} |

*(Row label: Intermediate Nodes Labeling)*

Table 7: Detailed prompt design for goal-plan trees refinement.

| Prompt |
| --- |
| #3 goal-plan Tress Refinement Stage – Given the goal-plan tree in JSON format, the AI assistant helps improve its rationality from two aspects: 1. Completing the missing internal steps, which can often be revised on commonsense (e.g., opening the refrigerator without closing it). 2. Enhancing the non-leaf node descriptions to be more abstract (e.g., from 'use toilet' to 'feel the call of nature'). Note that: You should only output the revised goal-plan tree in JSON. To facilitate goal-plan tree refinement, a set of illustrative cases is provided for reference: {{Demonstrations}}. |

| Demonstrations |
| --- |
| Please refine the goal-plan tree:
**Query:**
{"Root": "use toilet", "children": ["node1", "node2"], "interchangeable groups": []}
 {"node1": "<walk>, <toilet>", "children": [] }
 {"node2": "<sit>, <toilet>", "children": []}
**Response:**
{"Root": "feel the call of nature", "children": ["node1", "node2"], "interchangeable groups": []}
 {"node1": "<walk>, <toilet>", "children": [] }
 {"node2": "<sit>, <toilet>", "children": []} |
| Please refine2the goal-plan tree:
**Query:**
{"Root": "evening routine", "children": ["node1", "node2", "node3"], "interchangeable groups": ["group1", "group2"]}
 {"node1": "watch TV", "children": ["node1-1", "node1-2", "node1-3"] }
  {"node1-1": "<walk>", <couch>", "children": []}
  {"node1-2": "<sit>, <couch>", "children": []}
  {"node1-3": "<press>", <remote>", "children": []}
 {"node2": "have dinner", "children": ["node2-1", "node2-2", "node2-3", "node2-4", "node2-5", "node2-6"] }
  {"node2-1": "<walk>, <refrigerator>", "children": [] }
  {"node2-2": "<open>, <refrigerator>", "children": [] }
  {"node2-3": "<take>, <food in refrigerator>", "children": [] }
  {"node2-4": "<walk>, <dining chair>", "children": [] }
  {"node2-5": "<sit>, <dining chair>", "children": [] }
  {"node2-6": "<eat>, <food in refrigerator>", "children": [] }
 {"node3": "watch TV", "children": ["node3-1", "node3-2"] }
  {"node3-1": "<walk>, <bed>", "children": [] }
  {"node3-2": "<lie>, <bed>", "children": [] }
 {"group1": [{"node1"}, {"node2"}]}
 {"group2":[{"node3"}]}
**Response:**
{"Root": "engage in the rituals of dusk", "children": ["node1", "node2", "node3"], "interchangeable groups": ["group1", "group2"]}
 {"node1": "indulge in the visual leisure", "children": ["node1-1", "node1-2", "node1-3"] }
  {"node1-1": "<walk>, <couch>", "children": []}
  {"node1-2": "<sit>, <couch>", "children": []}
  {"node1-3": "<press>, <remote>", "children": []}
 {"node2": "partake in the evening nourishment", "children": ["node2-1", "node2-2", "node2-3", "node2-4", "node2-5", "node2-6"] }
  {"node2-1": "<walk>, <refrigerator>", "children": []}
  {"node2-2": "<open>, <refrigerator>", "children": []}
  {"node2-3": "<take>, <food in refrigerator>", "children": []}
  {"node2-4": "<walk>, <dining chair>", "children": []}
  {"node2-5": "<sit>, <dining chair>", "children": []}
  {"node2-6": "<eat>, <food in refrigerator>", "children": []}
 {"node3": "embrace the rituals preceding slumber", "children": ["node3-1", "node3-2"] }
  {"node3-1": "<walk>, <bed>", "children": []}
  {"node3-2": "<lie>, <bed>", "children": []}
 {"group1": [{"node1"}, {"node2"}]}
 {"group2":[{"node3"}]} |

*goal-plan Tress Refinement*

Table 8: Top 100 objects by frequency in BEHAVIORHUB dataset.

| Object List |
| --- |
| Pillow, Door, Lamp, Floor, Window, Cabinet, Box, Book, Chair, Shelf, Table, Mirror, Curtain, Towel, Paint, Bag, Shoe, Clothes, Sink, Bed, Stairs, Toy, Tap, Cardboard Box,Rug, Toilet, Beam, Basket, Armchair, Wall Lamp, Drawer, Decoration, Shower Wall, Pipe, Wardrobe, Vase, Toilet Paper, Picture, Cushion, Bottle, TV, Carpet, Desk, Decorative Plant, Radiator, Door Knob, Ventilation, Blanket, Hanger, Blinds, Couch, Photo, Clutter, Stool, Trashcan, Container, Window Curtain, Appliance, Ornament, Flowerpot, Product, Candle, Device, Storage Box, Rack, Refrigerator, Nightstand, Dining Chair, Light Fixture, Support Beam, Basket of Something, Curtain Rod, Towel Bar, Vent, Bathroom Cabinet, Plate, Speaker, Heater, Window Glass, Kitchen Appliance, Bathroom Accessory, Faucet, Kitchen Lower Cabinet, Clock, Flower Vase, Board, Hanging Clothes, Cabinet Door, Cup, Table Lamp, Dresser, Air Vent, Case, Cloth, Bathtub, Bin, Flower, Can, Bowl, Cosmetics |

Table 9: Top 50 actions by frequency in BEHAVIORHUB.

| Action List |
| --- |
| Walk, Sit, Stand Up, Move, Place, Open, Take, Clean, Jump, Run, Throw, Eat, Turn, Pick Up, Put On, Touch, Lift, Grasp, Dance, Knock, Yoga, Catch, Grab, Lie, Play, Shake, Hit, Drink, Stop, Give, Wash, Close, Relax, Remove, Rub, Check, Wait, Cut, Cook, Write, Tap, Press, Hang, Tie, Draw, Chop, Fill, Brush, Sleep, Flip |

Example JSON - Prepare for the day.

```
[
  {
    Root: prepare for the day
    children: [
      {
        node1: getting dressed
        children: [
          {
            node1-1: <open>, <wardrobe>
            children: []
          },
          {
            node1-2: <take>, <clothes>
            children: []
          },
          {
            node1-3: <put on>, <clothes>
            children: []
          }
        ]
      },
      {
        node2: breakfast preparation
        children: [
          {
            node2-1: <walk>, <kitchen>
            children: []
          },
          {
            node2-2: <open>, <refrigerator>
            children: []
          },
          {
            node2-3: <take>, <food in
              refrigerator>
            children: []
          },
          {
            node2-4: <walk>, <dining chair>
            children: []
          },
          {
            node2-5: <sit>, <dining chair>
            children: []
          },
          {
            node2-6: <eat>, <food in refrigerator
              >
            children: []
          }
        ]
      },
      {
        node3: home cleaning
        children: [
          {
            node3-1: <walk>, <bedroom>
            children: []
          },
          {
            node3-2: <take>, <broom>
            children: []
          },
          {
            node3-3: <clean>, <floor>
            children: []
          }
        ]
      },
      {
        node4: work from home
        children: [
          {
            node4-1: <walk>, <computer desk>
            children: []
          },
          {
            node4-2: <sit>, <computer chair>
            children: []
          },
          {
            node4-3: <typing>, <keyboard>
            children: []
          }
        ]
      }
    ],
    interchangeable groups: [
      {
        group1: [node1]
      },
      {
        group2: [node2, node3, node4]
      }
    ]
  }
]
```

Example JSON - Prepare dinner.

```
[
  {
    Root: prepare dinner
    children: [
      {
        node1: gather ingredients
        children: [
          {
            node1-1: <walk>, <refrigerator>
            children: []
          },
          {
            node1-2: <open>, <refrigerator>
            children: []
          },
          {
            node1-3: <take>, <food in
                refrigerator>
            children: []
          }
        ]
      },
      {
        node2: cook ingredients
        children: [
          {
            node2-1: <walk>, <stove>
            children: []
          },
          {
            node2-2: <place>, <food in
                refrigerator>, <stove>
            children: []
          },
          {
            node2-3: <wait>
            children: []
          },
          {
            node2-4: <check>, <food in
                refrigerator>
            children: []
          },
          {
            node2-5: <cut>, <food in refrigerator
                >
            children: []
          },
          {
            node2-6: <cook>, <food in
                refrigerator>
            children: []
          }
        ]
      },
      {
        node3: serve dinner
        children: [
          {
            node3-1: <walk>, <dining table>
            children: []
          },
          {
            node3-2: <sit>, <dining chair>
            children: []
          },
          {
            node3-3: <eat>, <food in refrigerator
                >
            children: []
          }
        ]
      }
    ],
    interchangeable groups: [
      {
        group1: [node1]
      },
      {
        group2: [node2]
      },
      {
        group3: [node3]
      }
    ]
  }
]
```

Example JSON - Weekend cleaning.

```
[
  {
    Root: Weekend cleaning
    children: [
      {
        node1: clean bedroom
        children: [
          {
            node1-1: <walk>, <bedroom>
            children: []
          },
          {
            node1-2: <take>, <broom>
            children: []
          },
          {
            node1-3: <clean>, <floor>
            children: []
          },
          {
            node1-4: <clean>, <window>
            children: []
          },
          {
            node1-5: wash clothes
            children: [
              {
                node1-5-1: <walk>, <washing
                    machine>
                children: []
              },
              {
                node1-5-2: <place>, <clothes>, <
                    washing machine>
                children: []
              }
            ]
          }
        ]
      },
                                                  {
                                                    node2: clean kitchen
                                                    children: [
                                                      {
                                                        node2-1: <walk>, <kitchen>
                                                        children: []
                                                      },
                                                      {
                                                        node2-2: <walk>, <dining table>
                                                        children: []
                                                      },
                                                      {
                                                        node2-3: <clean>, <dining table>
                                                        children: []
                                                      },
                                                      {
                                                        node2-4: wash dishes
                                                        children: [
                                                          {
                                                            node2-4-1: <walk>, <sink>
                                                            children: []
                                                          },
                                                          {
                                                            node2-4-2: <wash>, <kitchen
                                                                appliance>
                                                            children: []
                                                          }
                                                        ]
                                                      }
                                                    ]
                                                  }
                                                ],
                                                interchangeable groups: [
                                                  {
                                                    group1: [node1, node2]
                                                  }
                                                ]
                                              }
                                            ]
```

