# OpenReview forum: "Towards Human-like Virtual Beings: Simulating Human Behavior in 3D Scenes"
_ICLR.cc/2025/Conference — ICLR 2025 Conference Withdrawn Submission_

### Official Review · Reviewer_ECg4 · 2024-11-03

**Soundness:** 3
**Presentation:** 2
**Contribution:** 3
**Rating:** 5
**Confidence:** 3

**Summary:**

This paper introduces ACTOR, an agent designed to achieve high-level, long-horizon abstract goals within 3D household environments, guided by internal values akin to human motivations. ACTOR functions within a perceive-plan-act cycle, enhancing a scene-agnostic, ungrounded LLM controller through systematic goal decomposition and informed decision-making. This is accomplished by actively exploring the behavior space, generating activity options based on a hierarchical framework, and assessing these options through customizable value functions to determine the most appropriate subsequent actions. Additionally, the paper presents the BEHAVIORHUB dataset, which automates the alignment of motion resources with 3D scenes to facilitate informed generation. Comprehensive experiments demonstrate that ACTOR significantly outperforms established baselines, nearly doubling the overall success rate.

**Strengths:**

1. Propose a hierarchical structure for generating long-horizon motions. The agent should process perceived information, break down the goal into a series of activities, and create suitable action plans for each activity.
2. The planning method must operate in a dynamic and open environment. It should be aware of whether the target space is occupied, allowing the agent to make decisions about what to do next rather than simply following a predefined script.
3. Create the BEHAVIORHUB dataset, which will include a hierarchical decision tree and related motions. This dataset is essential for advancing more embodied tasks.

**Weaknesses:**

1. All the subgoals are generated by the LLM, but the results are not always reliable. How do you address any nonsensical outputs? Is there a significant amount of human feedback during the process?
2. The input to the LLM isn't very clear beyond just the object and action. Should we also consider global position, direction, volume of objects, and so on?
3. In the real-valued function, what does “distance” refer to?
4. The ACTOR uses a transformer-based algorithm to blend motions, but Figure 4 only shows different human-object interactions. It does not illustrate the connections and continuity between different actions, which contradicts one of the article’s contributions regarding long-horizon planning.

**Questions:**

see above Weakness

---

### Official Review · Reviewer_GLBa · 2024-11-04

**Soundness:** 2
**Presentation:** 3
**Contribution:** 2
**Rating:** 5
**Confidence:** 4

**Summary:**

This paper introduces ACTOR, a large language model-powered agent designed to simulate high-level, long-horizon, abstract goal-driven human behaviors in 3D scenes. The agent operates in a perceive-plan-act loop and utilizes a value-driven planning mechanism to navigate complex environments and adapt to dynamic situations. The authors compare the proposed MCTS based hierarchical framework with other LLMs to demonstrate its effiveness.

**Strengths:**

Comprehensive goal achievement: ACTOR can decompose high-level goals into a series of activities and actions, allowing it to accomplish complex tasks similar to humans.

Environmental awareness: The agent can adapt its plans based on environmental changes, such as the occupancy of a room or the state of objects.

Value-driven decision-making: ACTOR uses customizable value functions to evaluate and prioritize different action paths, incorporating personal beliefs and preferences.

Large-scale dataset: BEHAVIORHUB, a dataset of human behavior in 3D scenes, provides a valuable resource for agent development and evaluation, addressing the lack of a comprehensive testbed for this research area.

**Weaknesses:**

The agent’s interaction with objects is static, and the agent’s actions do not directly impact the state of the environment, which makes it only a benckmark for LLMs and cannot further applied to multi-modal policies.

The compared baselines are outdated, more existing planning frameworks that utilized CoT, ToT should be introduced.

The design of the environment state is too simple and does not consider multi-human interaction, which makes the frameworks hard to deploy in real world.

**Questions:**

See weakness.

---

### Official Review · Reviewer_nfxc · 2024-11-04

**Soundness:** 2
**Presentation:** 2
**Contribution:** 1
**Rating:** 3
**Confidence:** 3

**Summary:**

The paper introduces ACTOR and BehaviorHub.
1) ACTOR is an agent architecture to produce human behavior data in 3D environments with ‘values’ similar to those humans set in daily life. The agent uses a perceive-plan-act cycle for scene-aware and goal-oriented planning; it also decomposes goals and decision making into a hierarchical space.

2) BehaviorHub is a human behavior simulation dataset with commonsense knowledge of LLMs and motion resources with 3D scenes.

**Strengths:**

The idea of grounding the dataset of human behaviors in 3D scenes might be useful for dataset diversity.

**Weaknesses:**

There are many design choices in ACTOR that were left unjustified. Additionally, since the evaluation dataset is not explained, it is difficult to gauge the performance of the method.

1) Dataset creation approach based on ‘values similar to those humans set in daily life’. This is difficult to evaluate when the 'ground-truth plans' mentioned in the paper are not explained e.g., in the example below
*For behavior planning, Sentence-BLEU (Papineni et al., 2002), and BERTScore (Zhang et al., 2019) are used to measure the semantic similarity between the ground-truth plans and predictions.* (5.4)

2) Introducing hierarchical decomposition of goals and decision making. Why should candidates at the same level be restricted to executable actions or high-level semantic units of activities? I could see multiple potential paths where it might be beneficial for some paths to be longer than others.

3) Perception module: Why does the below prompt *ENVIRONMENT: {residential interior}; OBJECTS: {bed, desk, chair, kitchen counter, sink, television,..., sofa}; SURROUNDINGS: {sink: empty, faucet: turned on, toilet: vacant}* help attain ‘deep understanding of the environment’.
Are there any experiments/ previous works to justify this structure of inputs?

**Questions:**

Clarification from weakness 1 would be helpful.

---

### Official Review · Reviewer_U9K1 · 2024-11-04

**Soundness:** 2
**Presentation:** 2
**Contribution:** 2
**Rating:** 5
**Confidence:** 4

**Summary:**

This paper presents ACTOR, an LLM system for simulating human-like behavior in 3D environments. It also proposes a comprehensive dataset, BehaviorHub, for training and evaluating such systems. ACTOR operates on a perceive-plan-act cycle, using value-based behavioral planning and hierarchical prior knowledge to decompose complex goals into achievable steps while adapting to environmental changes. The BehaviorHub dataset contains 10k human behavior samples in 1.5k 3D scenes, generated by a semi-automated pipeline that uses language models to generate plausible behavior sequences with corresponding motion data. The paper also presents an evaluation framework that measures both behavior planning and motion simulation effectiveness, and demonstrates ACTOR's superior performance compared to existing approaches.

**Strengths:**

The teaser figure of this paper clearly illustrates the key differences of this paper compared to the prior art. Generating long-horizon human behaviors from high-level abstract description is promising in many fields. This paper provides a comprehensive evaluation of their proposed method, including both quantitative metrics and human evaluation, supported by thorough ablation studies.

**Weaknesses:**

There are a few notable issues that concern me.

1. Although the generation of human-like behavior from high-level description is promising, the proposed method does not show sufficient technical contributions to shed light on this problem. The use of LLM for manipulating agents or robots is not new (see [a, b]). Adopting the perception-decision-action loop is straightforward and not new either (see [c, d]). I do not see a clear statement that articulates the core contribution in techniques compared to previous methods.

2. The value functions used seem relatively simplistic, using basic metrics such as shortest path and binary language-based decisions, rather than incorporating more sophisticated physical constraints or human behavior patterns. The probabilistic formulation of the value function used in this paper is somewhat trivial, as it is simply implemented by constructed decrete values, i.e., 1.0/0.7/0.3/0.01. Such a probabilistic model (p_v) does not involve any learning process, how to guarantee the that the used distribution match the actual underlying distribution?

3. In addition, the baselines (LLMaP and HuggingGPT) aren't specifically designed for 3D human behavior simulation, and the human evaluation sample size is small with only five participants. It would be better if some more task-oriented baselines were used in the experiments.

[a] J. Liang, W. Huang, F. Xia, P. Xu, K. Hausman, B. Ichter, P. Florence, and A. Zeng, "Code as policies: Language model programs for embodied control," in International Conference on Robotics and Automation (ICRA), 2023.

[b] I. Singh, V. Blukis, A. Mousavian, A. Goyal, D. Xu, J. Tremblay, D. Fox, J. Thomason, and A. Garg, "Progprompt: Generating situated robot task plans using large language models," in International Conference on Robotics and Automation (ICRA), pp. 11523-11530, IEEE, 2011.

[c] Y. Hu, F. Lin, T. Zhang, L. Yi, and Y. Gao, "Look before you leap: Unveiling the power of gpt-4v in robotic vision-language planning," arXiv preprint arXiv:2311.17842, 2023

[d] M. Skreta, Z. Zhou, J. L. Yuan, K. Darvish, A. Aspuru-Guzik, and A. Garg, "Replan:

**Questions:**

1. How do you model the belief in the value function. This part is not well explained in the paper.
2. How did you select and calibrate the probability values (1.0/0.7/0.3/0.01) for language-based commands?

---

### Note · Authors · 2024-11-15

I have read and agree with the venue's withdrawal policy on behalf of myself and my co-authors.